# Molecular Biomarkers in Fragile X Syndrome

**DOI:** 10.3390/brainsci9050096

**Published:** 2019-04-27

**Authors:** Marwa Zafarullah, Flora Tassone

**Affiliations:** 1Department of Biochemistry and Molecular Medicine, University of California Davis, School of Medicine, Sacramento, CA 95817, USA; mzafarullah@ucdavis.edu; 2MIND Institute, University of California Davis Medical Center, Sacramento, CA 95817, USA

**Keywords:** fragile X syndrome, molecular biomarkers, *FMR1*, FMRP, intellectual disability, *Fmr1* KO mouse, ASD

## Abstract

Fragile X syndrome (FXS) is the most common inherited form of intellectual disability (ID) and a known monogenic cause of autism spectrum disorder (ASD). It is a trinucleotide repeat disorder, in which more than 200 CGG repeats in the 5’ untranslated region (UTR) of the fragile X mental retardation 1 (*FMR1*) gene causes methylation of the promoter with consequent silencing of the gene, ultimately leading to the loss of the encoded fragile X mental retardation 1 protein, FMRP. FMRP is an RNA binding protein that plays a primary role as a repressor of translation of various mRNAs, many of which are involved in the maintenance and development of neuronal synaptic function and plasticity. In addition to intellectual disability, patients with FXS face several behavioral challenges, including anxiety, hyperactivity, seizures, repetitive behavior, and problems with executive and language performance. Currently, there is no cure or approved medication for the treatment of the underlying causes of FXS, but in the past few years, our knowledge about the proteins and pathways that are dysregulated by the loss of FMRP has increased, leading to clinical trials and to the path of developing molecular biomarkers for identifying potential targets for therapies. In this paper, we review candidate molecular biomarkers that have been identified in preclinical studies in the FXS mouse animal model and are now under validation for human applications or have already made their way to clinical trials.

## 1. Introduction

A biomarker is “a characteristic that is objectively measured and evaluated as an indicator of normal biological processes, pathogenic processes, or pharmacologic responses to a therapeutic intervention” [1]. Biomarkers can be found in blood, plasma, or other tissues and are generally viewed as a molecular signature able to identify individuals who are at high risk for a specific condition. They can also be detected before disease symptoms and therefore used to predict the occurrence of a condition or the nature and severity of disease outcomes in an individual. Importantly, they can be used to evaluate the efficacy of response to pharmacological intervention.

Fragile X syndrome (FXS) is the most prevalent inherited cause of intellectual disability and the single leading monogenic known cause of autism, as 60% of those with a full mutation present with autism spectrum disorder (ASD) [2]. The clinical symptoms include anxiety, impairment in cognitive, executive and language performance, hyperactivity, impulsivity, insomnia, seizures and physical features such as hypotonia, flat feet, hyperextensible joints, and macroorchidism [3]. FXS is caused by the abnormal expansion, greater than 200 units of a naturally occurring CGG repeat in the 5’ untranslated region (UTR) of the fragile X mental retardation 1 (*FMR1*) gene, located on the X chromosome. This expansion, named full mutation, results in hypermethylation and transcriptional silencing of the gene, leading to the loss or reduction of fragile X mental retardation 1 protein (FMRP) expression and to the diagnosis of fragile X syndrome [4,5,6]. Individuals carrying expansion of 55–200 CGG repeat are premutation carriers and at risk of developing the late-onset neurodegenerative syndrome, fragile X-associated tremor/ataxia syndrome (FXTAS), the fragile X-associated primary ovarian insufficiency (FXPOI) [7] and the fragile X-associated neuropsychiatric disorders (FXAND) [8].

FMRP is an RNA-binding protein and a translational regulator, whose function affects synaptic plasticity, spine morphology, and several cellular signaling pathways. Reduced expression of FMRP leads to the abnormalities in neurodevelopmental processes and the disturbed neuronal communications observed in FXS [9]. Young adults and adolescents with FXS show neuroanatomical abnormalities [10], and the regions of the brain that are significantly impacted by the loss of FMRP are the hippocampus (a structure that plays a critical role in the learning and memory and the regulation of mood and cognition [11]), the cerebellum, and the basal forebrain (nucleus basalis) [12]. Several studies in the *Fmr1* knockout (KO) mouse model suggest that FMRP plays a critical role during specific periods of cortical development with regional brain volume changes occurring in adult mouse brain [13,14]. Brain volume changes have also been observed in children with FXS, specifically in the temporal lobe, cerebellum, caudate nucleus, and amygdala regions of the brain [15,16].

FMRP function appears to be mostly inhibitory as it prevents the activity of various biochemical pathways in a “controlled” manner [17]. In a sense, reduced FMRP leads to exaggerated or reduced biochemical reactions that can adversely affect neural function. The past two decades of research have shown defects in the central excitatory glutamatergic and inhibitory GABAergic pathways and in several other neurotransmitter systems including serotonin and dopamine [18,19]. Thus, the development of molecular measures that reflect the impact of a drug on one or more of the FMRP-regulated pathways (Figure 1), including the activity or the expression level of proteins in the translational activation pathway and particularly of those regulated by FMRP, could potentially act as molecular biomarkers for FXS.

The *Fmr1* knockout (KO) mouse model [20], lacks a functional *FMR1* gene and therefore does not express FMRP. Many studies have shown that the *Fmr1* KO mouse presents with some phenotypes that resemble the human disorder, including biochemistry [21], electrophysiology [22], neuropathology [23], and spine morphology [24]. Although the observed patterns of brain activity, including audiogenic seizures, are similar to those in individuals affected by FXS [25], these mice poorly mimic human behavior. Indeed, the strains of the *Fmr1* KO mouse that are often used to test drugs for FXS do not show the cognitive problems seen in patients with FXS [26]. Nevertheless, a large body of literature on the *Fmr1* KO mouse has paved the way to preclinical studies which have shown to rescue several of the FXS phenotypes [27] and have ultimately led to clinical trials in patients with FXS.

Hope has been tempered by the lack of translating the positive results observed in the *Fmr1* KO mouse model into therapy in a clinical setting. Currently, nonpharmacological and behavioral treatments are symptomatic, and they can be coupled with pharmacological treatments of anxiety, aggression, and attention deficit hyperactivity disorder (ADHD).

To date, there is no cure for FXS, and the recent failures of multiple clinical trials have highlighted the need for the development and validation of new biomarkers to better measure the clinical outcome of these treatments [28,29]. Many studies aimed to a better understanding of the underlying molecular mechanisms and pathways involved in FXS have led to the development of specific biomarkers for defining targeted therapeutic strategies intended to reverse the intellectual and behavioral problems of patients with FXS. In this paper, we will review the proposed candidate molecular biomarkers (Figure 2) that have been identified in *Fmr1* KO mouse as an early sign of drug promise and in some cases, later moved to a clinical trial in patients with FXS.

## 2. *FMR1* Molecular Measures

*FMR1*-related measures, including CGG repeat number, percent of methylation, *FMR1* mRNA and FMRP expression levels have been correlated to neurocognitive and social–affective functioning assessments and mental health problems in individuals with FXS [30,31,32,33,34,35,36,37,38]. The magnitude of the observed correlations generally suggests that these molecular biomarkers are likely accounting only for a proportion of the phenotypic variability of this disorder.

Variation in CGG repeat size and methylation, so-called mosaicism, could be a useful biomarker of various types of risks that could affect subjects with FXS. Mosaicism defines differences in gene expression between those with fully hypermethylated *FMR1* alleles and those carrying unmethylated alleles, and ultimately reflects the levels of expression of *FMR1* mRNA and FMRP. Generally, mosaicism refers to the presence of a full mutation allele(s) and a premutation allele (size mosaicism) or the presence of a full methylated allele(s) and unmethylated alleles (methylation mosaicism), throughout the CGG repeat size range.

Sex differences undoubtedly contribute to the severity of the FXS phenotype; indeed, intellectual and developmental disability is observed in 85% of males and only in 25% of females [27,39,40]. In females, who have two X chromosomes, the process of X inactivation, early during embryonic development, leads to methylation and therefore inactivity of one X chromosome in each cell. However, due to the presence of the chromosome carrying the normal allele, the impact of the *FMR1* mutation in females is reduced relative to males, who have only one X chromosome [41]. The relative proportion of the normal allele on the active and inactive X chromosomes, so-called activation ratio (AR), has shown to contribute to differences in affectedness among females, making the AR a useful biomarker for determining the severity of the phenotype. It should be noted that since X inactivation is a random process, it could be different in different tissues, such as blood and brain [42,43,44].

## 3. Metabotropic Glutamate Receptors (mGluRs)

The “mGluR theory of FXS” states that the absence of FMRP leads to excessive metabotropic glutamate receptors (mGluRs, mGluR1 and mGluR5) activated long-term depression (LTD) and reduced responsiveness to signals in the hippocampus and other parts of the brain involved in memory and learning. Together, they are contributing to the neurological and psychiatric symptoms of FXS [45,46,47]. Reduction of mGluR signaling has demonstrated a reversal of the fragile X phenotypes providing substantial support to the involvement of the mGluR5 pathway in FXS [48]. For more than a decade, our understanding of the molecular pathophysiology of FXS has been substantially advanced by the corroboration of “mGluR theory of FXS” in a wide range of experiments with a number of different mGluR5 inhibitors tested in both the *Fmr1* KO mouse [49,50,51,52,53] and in the Drosophila models of FXS [54,55,56,57,58,59,60,61,62]. *Fmr1* mutant mouse with a 50% reduction in mGluR5 expression was generated to demonstrate that a range of FXS phenotypes could be corrected by downregulating signaling through group 1 mGluRs [45]. Their findings showed that the decrease in mGlu5 expression levels from early embryonic development effectively prevented the onset of a broad range of FXS phenotypes, including audiogenic seizures, increased basal protein synthesis, spine density, although no effect on macroorchidism was observed.

MPEP (2-methyl-6-phenylethynyl-pyridine) was the first mGluR5-antagonist tested in the *Fmr1* KO mouse, which demonstrated rescue of behavioral defects, including open field performance [63], the rescue of the spine/filopodia ratio in *Fmr1* KO neurons to the levels observed in wild-type neurons [64]. Further, MPEP treated *Fmr1* KO mouse showed improved behavior by significantly fewer errors, less perseveration, and impulsivity when navigating mazes, in addition to reverse postsynaptic density-95 (PSD-95) protein deficits which, if confirmed, could be considered a molecular biomarker [65]. Finally, MPEP prevented an abnormal clustering of DHPG (group I mGluR agonist (S)-3,5-dihydroxyphenylglycine) responsive cells (responsible for activation of ionotropic receptors in mouse FXS neurospheres) and corrected morphological defects of differentiated cells [66].

Similarly, a study on chronic treatment of *Fmr1* KO mouse with the long-acting mGlu5 inhibitor 2-chloro-4-((2,5-dimethyl-1-(4-(trifluoromethoxy)phenyl)-1H-imidazol-4-yl)ethynyl) pyridine (CTEP), fully corrected numerous phenotypes including the increased synaptic spine density, protein synthesis rate, aberrant synaptic plasticity, learning and memory deficits, increased body growth rate, and sensitivity to audiogenic seizures. In addition, this study shows a reduction of both extracellular signal-regulated kinase (ERK) activity and mTOR phosphorylation levels in the *Fmr1* KO but not in wild-type (WT) animals, suggesting that they could represent potential biomarkers in FXS [53]. These studies have shown that the long-term, uninterrupted mGluR5 inhibition is essential for a successful pharmacological intervention as a single dose of the mGluR5 inhibitors was not sufficient to correct the mouse phenotypes. [50,53]. One of the potential molecular mechanisms for mGluR5 dysfunction in FXS is the decreased association of mGluR5 with the Homer family of scaffolding proteins. Indeed, genetic deletion of H1 (an activity-inducible isoform of Homer1) restored regular mGluR5-long Homer association in the *Fmr1* KO and corrected much of the mGluR5 dysfunction as well as behavioral phenotypes, including anxiety and audiogenic seizures [67]. Further, the disruption of mGluR5-Homer resulted in phenotypes of FXS including reduced mGluR5 association with the postsynaptic density, deficits in agonist-induced translational control, protein synthesis-independent LTD, neocortical hyperexcitability, audiogenic seizures, and altered behaviors, such as anxiety and sensorimotor gating [68].

The Drosophila genome encodes only a single mGluR (DmGluRA), compared to the eight separate receptors in mammals [69]. The simplicity of the Drosophila system, coupled with the evolutionary conservation of the activation pathways, has provided an excellent model to test the mGluR hypothesis. Treatment with lithium and MPEP restored normal courtship behavior, mushroom, body defects, and short-term memory, but not β-lobe crossing, suggesting that other morphological abnormalities are responsible for the memory defects [54,70].

Molecular analyses reveal an inverse relationship between dFMRP and DmGluRA, with the latter overexpressed in *dFmr1* null animals and dFMRP overexpressed in DmGluRA nulls [57]. The DmGluRA null also shows more striking defects in activity-dependent synaptic function, including high transmission amplitudes during high-frequency stimulation and abnormally strong hyper potentiation following high-frequency stimulation [57,58]. The successful unbiased screen for small molecules that can rescue the lethality of glutamate-treated larvae and adults *dFmr1* mutants, using the mGluR5 noncompetitive antagonist MPEP or LiCl has been reported to rescue naïve courtship behavior, immediate recall memory, and short-term memory of *dFmr1* mutants [56,59]. The compelling results of these preclinical studies, showing evidence of benefits in rodent and Drosophila disease models, have prompted the application of mGlu5 inhibitors as potential target treatments in human clinical trials for FXS. Thus, clinical trials in FXS patients have been conducted to explore the safety, tolerability, and efficacy of a number of different mGluR5 antagonists.

Fenobam [71], the first mGlur5 antagonist drug evaluated in a single-dose open-label study of 12 male and female adults with FXS (mean age 23.9 years), showed trends of improvement in a prepulse inhibition deficit relative to controls who did not receive the drug [72]. Subsequently, in an exploratory study, the efficacy of mavoglurant (AFQ056) [73] was tested in a randomized double-blind crossover study of 30 FXS males. In this study, seven patients with a hypermethylated full mutation with no detectable *FMR1* mRNA expression, improved stereotypic behavior, hyperactivity, and inappropriate speech, while no improvement found in 18 patients with partial promoter methylation [74]. Thus, it appears that those with full methylation responded best, whereas those who were mosaics with partial methylation had a variable response with a lack of overall efficacy in that group. Although methylation is often regarded as a biomarker, results to date do not explain why some of those with lack of methylation responded and others did not [74]. In addition, the reported behavioral effects of stereotypic behavior, hyperactivity, and inappropriate speech were not replicated with FXS male and female adolescents and adults either full or partial *FMR1* methylation in subsequent 12-week double-blind mavoglurant studies [75].

Similarly, extensive proof of concept study was conducted with basimglurant, a potent and selective mGluR5-negative allosteric modulator (NAM) [76,77] and mavoglurant in male and female adults with FXS. In spite of their promising results in preclinical studies [77,78,79,80] these studies ended because no improvement in the clinical phenotype of patients enrolled in the clinical trials using these modulators were observed [29,81,82]. Recently, in a phase 2 12-weeks double-blind clinical trial, basimglurant did not demonstrate improvement over placebo in a parallel-group study of 183 adults and adolescents (aged 14–50, mean 23.4 years) with FXS [83]. Later, the study reported the long-term safety and efficacy of mavoglurant in the two open-label extensions in adolescent (*n* = 119, aged 12–19 years) and adult (*n* = 148, aged 18–45 years). In both studies, mavoglurant was well tolerated, and moderate behavioral improvements were observed in FXS as compared to the placebo control group. Thus, the compelling preclinical evidence for the therapeutic potential of mGlu5 inhibitors in the mouse and the Drosophila disease models has not translated in the anticipated benefits and improvement of the phenotype in FXS patients [84].

## 4. γ-Aminobutyric Acid (GABA) Receptors

GABA is the most prominent inhibitory neurotransmitter that acts through three receptors in the brain. GABA_A_ receptors are ligand-regulated chloride channels that upon activation cause hyperpolarization in mature neurons; GABA_B_ receptors are heterodimeric G protein-coupled receptors (GPCRs) which are mostly expressed presynaptically in the brain; and, GABA_C_ is CYS-loop ligand-gated ion channels receptors with a similar pentameric structure to GABA_A_ but are homomeric. FMRP directly binds several GABA_A_ receptor (α1, α2, α3, δ, and γ2) mRNAs of which expression is reduced in the cortex and cerebellum of young *Fmr1* KO mouse. Thus, the mRNA expression level of these subunits could be used as biomarker; however, they have not been studied in clinical trials for FXS with any GABA agonists [85]. The introduction of a yeast artificial chromosome (YAC) containing the “healthy” human *FMR1* genomic region into *Fmr1* KO mouse rescued the expression of these specific subunits of GABA_A_ receptors [86]. A recent electrophysiological study supported the notion that the δ subunit of the aminobutyric acid type A receptors (GABA_A_Rs) is compromised in the *Fmr1* KO mouse, by reporting a 4-fold decrease in tonic inhibition [87].

The delay in switching from depolarizing to hyperpolarizing GABA has also been observed in the cortex of *Fmr1* KO mouse during development [88]. Moreover, the oxytocin-mediated, GABA excitation–inhibition shift that occurs in newborn rodents during delivery is absent from the hippocampal neurons of *Fmr1* KO mouse. As a result, the hippocampal neurons have elevated intracellular chloride levels, increased excitatory GABA, enhanced the glutamatergic activity, and elevated gamma oscillations [89].

In a study, the response of the FXS neurons (differentiated in vitro from human embryonic stem cells lacking synaptic activity) has been investigated by pulse application of the neurotransmitter GABA. The results confirmed that human FXS neurons do not respond to GABA as FMRP plays a role in the development of the GABAergic synapse during neurogenesis, and that might be one of the potential reasons of the observed default synaptic activity in FXS patients. [90]. Some GABA agonists have been used in the *Fmr1* KO mouse to rescue behavioral abnormalities. The primary neuron excitability deficits in the amygdala of the *Fmr1* KO mouse was restored by gaboxadol (THIP), a GABA_A_ receptor agonist, which also improved some specific behavioral characteristics, including hyperactivity and auditory seizures [91]. The treatment of the *Fmr1* KO mouse with bumetanide (specific NKCC1 chloride importer antagonist) normalized electrophysiological abnormalities in the mutant offspring as well as hyperactivity and autistic behaviors [89]. Finally, arbaclofen, a GABA_B_ agonist, improved protein synthesis, the abnormal auditory-evoked gamma oscillations, working memory and anxiety-related behavior in *Fmr1* KO mouse [92,93,94].

Thus, these findings from different studies in the FXS animal models confirmed that GABA receptors are suitable targets for target treatment in FXS [18,39,95,96,97,98,99,100,101,102,103]. Indeed, two phase 3 placebo-controlled trials were conducted (with subjects aged 12–50 and in subjects aged 5–11) to determine the safety and efficacy of arbaclofen for improving social behavior in FXS patients. Although, arbaclofen did not meet the primary outcome measures of improved social avoidance in FXS in either study [104], in a double-blind placebo-controlled crossover trial [105], improved social function and behavior were reported in FXS patients. Acamprosate, which activates GABA_B_ and GABA_A_ receptors, also improved several phenotypes like cortical upstate duration, behavioral improvement, anxiety, locomotor tests in *Fmr1* KO mouse and reduced ERK1/2 activation in brain tissue [106]. Acamprosate has also been tested in an open-label 10-week trial of 12 young children aged 6–17 years with FXS. It was found safe and well-tolerated and resulted in better social behavior and reduced hyperactivity [107]. Ganaxolone is a neurosteroid and a positive GABA_A_ modulator that rescued several phenotypes in the *Fmr1* KO mouse, like increased marble-burying assay, sensory and sensorimotor gating in the acoustic startle response, and prepulse inhibition [86]. Tested in a recent randomized double-blind placebo-controlled crossover trial in children with FXS, aged 6–17, years, ganaxolone was found to be safe and have beneficial effects in some patients, particularly for those with higher anxiety or lower cognitive abilities [108]. These preclinical and clinical studies strengthen the hypothesis of GABA receptors involved in the pathology of FXS and as they are the major inhibitory receptors in the brain, they point to the therapeutic potential of the GABA receptor particularly for the behavioral and epileptic phenotypes associated with fragile X syndrome.

## 5. De Novo Protein Synthesis

Synaptic strength plays a crucial role in learning and memory and it is compromised in many neurodevelopmental disorders. One of the molecular mechanisms that regulate spine morphology, and therefore synaptic strength, is local de novo protein synthesis that enables synapses to alter their function and structure autonomously [109]. FMRP, an RNA binding protein which acts as a translational repressor of many synaptic proteins, is crucial in regulating this process, and the partial or complete lack of FMRP in FXS leads to increased protein translation at the synapses. The metabotropic glutamate receptor subtype 5 (mGluR5) theory of FXS also indicate that the imbalance of mechanisms involved in synaptic shaping and protein translation are responsible for many of the symptoms observed in FXS patients [49]. The lack of FMRP also leads to a loss of translational control and to increased rates of cerebral protein synthesis (rCPS) in some regions of the brain including the hippocampus, thalamus, and hypothalamus of the *Fmr1* KO mouse model of FXS [110].

Fibroblasts from FXS patients also showed significantly elevated rates of basal protein synthesis along with increased levels of the phosphorylated target of rapamycin (p-mTOR), phosphorylated extracellular signal-regulated kinase ½ (ERK1/2), and phosphorylated p70 ribosomal S6 kinase 1 (p-S6K1) [111]. Similarly, a recent study reported that the level of protein synthesis increased in fibroblast cell lines derived from individuals with FXS and from *Fmr1* KO mouse. However, this cellular phenotype displayed a broad distribution with a proportion of individuals with FXS and in the *Fmr1* KO mouse, showing a basal de novo protein synthesis within the normal range. These findings indicate that the molecular mechanisms that control protein synthesis are the primary targets in FXS. However, altered protein synthesis may not be the cause of all symptoms observed in FXS and, therefore, those with normal levels of protein synthesis are not likely going to benefit from target treatments aimed to lower protein synthesis [112]. Thus, de novo protein synthesis could be a very useful biomarker to predict phenotypic subgroups, symptoms severity, and treatment response. Further, as the treatment of fibroblast cells derived from FXS patients, with small molecules that block S6K1 and phosphoinositide 3-kinase (PI3K) catalytic subunit p110β, decreased the rates of protein synthesis in both control and patient fibroblasts; the role of these targets as a potential biomarker should be considered [111]. FXS subjects, under propofol sedation, showed a reduced rCPS in whole brain, cerebellum, and cortex compared to sedated controls. Similar results have been observed in most regions examined in sedative *Fmr1* KO mouse as compared to the WT mouse suggesting that changes in synaptic signaling can correct increased rCPS in FXS [113]. Chronic dietary lithium treatment also demonstrated to be efficacious in reversing the increased rCPS in the *Fmr1* KO mouse [114].

Some studies have shown that the mechanisms regulating the levels of protein synthesis, can be restored by reducing the mGluR5 signaling genetically or with pharmacological treatments [46,53,100,115,116,117,118]. Moreover, haploinsufficiency of mGluR5, reduction of MMP9, of striatal-enriched tyrosine phosphatase (STEP) signaling, or of S6K signaling can not only restore the levels of protein synthesis but also restore the synaptic and behavioral phenotypes in the FXS mouse model [50,119,120,121,122,123,124,125,126]. Recently, a study showed that treatment of the *Fmr1* KO mouse with a cell-permeable peptide able to modulate ADAM metallopeptidase domain 10 (ADAM10) activity and amyloid-β protein precursor (APP) processing, restored protein synthesis to the wild-type (WT) level [127].

These preclinical and clinical studies suggest that basal protein synthesis could be considered as a potential biomarker and a molecular hallmark for FXS, but unfortunately, replicating this optimal translational scenario into reality has not been fully successful [27]. The extent to which excessive protein synthesis associated with cognitive and behavioral impairments also remained unknown. More importantly, none of the human studies have shown an effect on the primary outcome measures which were mainly behavioral questionnaires in children, adolescents, or adults with FXS [74,104,105]. Finally, although FMRP modulates protein synthesis, there are other factors (environmental and genetic) that may contribute to the modulation of homeostasis of molecules involved in synaptic plasticity.

## 6. Phosphoinositide 3-Kinase (PI3K)

Phosphoinositide 3-kinase (PI3K) is the signaling molecule involved in cell motility, survival, growth, and proliferation. PI3K class I catalytic subunits, p110α, p110β, p110γ, and p110δ, have their specific dysregulation in FXS [128]. FMRP regulates the synthesis and synaptic localization of p110β, which is a crucial signaling molecule downstream of group 1 metabotropic glutamate receptor (gp1 mGluRs) and other membrane receptors. Lack of FMRP in the *Fmr1* KO mouse leads to excess mRNA translation and synaptic protein expression of p110β [123]. Treatment with a p110β-selective antagonist was effective in rescuing the excess of protein synthesis in the *Fmr1* KO mouse synaptoneurosomes and in lymphoblastoid cells derived from FXS patients [123,129]. Further, a prefrontal cortex (PFC) selective knockdown of p110β, reversed deficits in higher cognition, normalized excessive PI3K activity, restored stimulus-induced protein synthesis, and corrected increased dendritic spine density in the *Fmr1* KO mouse [130,131]. Thus, PI3K activity in patient cells might be a biomarker and could be used to assess the efficacy of drug response in target treatment in FXS.

## 7. Mammalian Target of Rapamycin (mTOR) and Substrate p70 Ribosomal S6 Kinase (S6K1)

Mammalian target of rapamycin (mTOR) is a 289 kDa serine/threonine kinase protein that controls various energetic functions at both the cellular and organism level and an essential regulator of cell proliferation, autophagy, translation, and growth. In neuronal cells, protein synthesis plays a fundamental role in the regulation of lasting alterations in synaptic strength or plasticity, and of long-term potentiation (LTP), processes that are important in learning and memory [132,133]. The components of the mTOR signaling cascade, which is involved in protein synthesis-dependent phase of synaptic strengthening, are present in dendrites suggesting a role for mTOR in local translation and synaptic plasticity. mTOR is activated in dendrites by stimulation of group I mGluRs and it is required for mGluR-LTD [134,135]. It has been reported that increased activity in these systems can lead to repetitive and perseverative behavior patterns [132].

The best-characterized function of mTOR is the regulation of translation. mTOR regulates two critical and core components of the translational initiation machinery, p70 ribosomal S6 kinase 1 and 2 (S6K1/2), and the eIF4E-binding proteins (4E-BPs), and it is also known to regulate the activity of phosphatases such as protein phosphatase 2A (PP2A). These phosphatases, in turn, regulate mTOR substrates, thereby generating mTOR-dependent feedback loops that control initiation rates. Increased phosphorylation of (mTOR) substrate, p70 ribosomal subunit 6 kinase 1 (S6K1) along with the high expression of mTOR regulator, and the serine/threonine protein kinase (Akt) was also observed in lymphocytes and brain tissues derived from subjects with FXS [136].

The enhanced mTOR signaling observed in the hippocampus of the *Fmr1* KO mouse associates with the increase eukaryotic initiation factor complex F4 (eIF4F) [137] and with the increased phosphorylation of the cap-binding protein eukaryotic initiation factor 4E (eIF4E) [136] to further support the increased protein synthesis observed in FXS. These findings, in both FXS mice and humans, are consistent with the idea that the loss of FMRP results in the dysregulation of mechanisms of translational initiation control rather than transcriptional regulation and provide the direct evidence that mTOR dysregulation may be useful for designing targeted treatments in FXS [136]. Therefore, targets and substrates in the mTOR signaling pathways can act as potential molecular biomarkers. Since the molecular signaling effects resulting from FMRP loss are likely causal in the wide-range of the severity of the FXS symptoms, including autism, identifying the effects of FMRP loss on molecular signaling pathways, like those governing translation, is key to advancing our ability to treat the disorder.

Finally, metformin, a type 2 diabetes medication that can improve obesity and excessive appetite, has emerged as a candidate drug for targeted treatment of FXS based on preclinical studies. These studies have shown rescue of a number of FXS phenotypes including memory deficits, social novelty, grooming, dendritic spine morphology, and electrophysiology in the CA1 of the hippocampus [138,139]. Metformin suppresses mRNA translation via inhibition of ERK and mTOR pathways, which are overactive in FXS, supporting their potential role as molecular biomarkers, and therefore, may contribute to normalizing signaling pathways in the CNS of FXS patients. In humans, metformin has been used in the clinical treatment of several individuals with FXS and showed benefits not only in lowering weight gain but also in improving language and behavior [138]. Thus, metformin shows promises for targeting several signaling pathways disrupted in FXS and possibly rescuing some of the clinical symptoms observed in individuals with FXS. Interestingly, a double-blind placebo-controlled trial of metformin in individuals with FXS is currently ongoing which will assess safety and benefit of metformin in the treatment of language deficits, behavioral problems, and obesity in individuals with FXS.

## 8. Extracellular-Regulated Kinase (ERK)

The ERK pathway is a chain of proteins in the cell that acts as a nodal point for cell signaling cascades. The absence of FMRP in *Fmr1* KO mouse results in rapid dephosphorylation of ERK upon mGluR1/5 stimulation suggesting that over-activation of phosphatases in synapses affects the synaptic translation, transcription, and synaptic receptor regulation in FXS [53,119,140,141]. Delayed early-phase phosphorylation of ERK is observed in both neurons and thymocytes of the *Fmr1* KO mouse. Likewise, the early-phase kinetics of ERK activation in lymphocytes from human peripheral blood is also delayed in individuals with FXS, as compared to controls [142]. The correction of the delayed ERK activation time, resulting in a faster activation, was observed after 2 months of treatment with lithium in a pilot open-label trial in FXS or with riluzole treatment [143,144]. These findings, based on a small number of subjects, suggest ERK activity as a potential biomarker for measuring the metabolic status of the disease in FXS.

Recently, the significant FMRP-dependent over-activation of ERK was observed in both FXS mouse and humans. ERK activity was normalized in FXS platelets [145], and correlated with clinical response to lovastatin, pointing this inhibitor of ERK pathway signaling cascade as a promising treatment for FXS [146]. The findings by Pellerin et al. [145] suggest that the use of platelet’s ERK activity represents a new potentially interesting biomarker for future clinical trials. It may also pave the way for other promising and very exciting discoveries that will eventually improve FXS patients’ assessment in future clinical trials where either lovastatin or other ERK-targeting drugs is applied.

## 9. Matrix Metalloproteinase-9 (MMP-9)

FMRP deficit is associated with alterations in the expression of a number of proteins, including matrix metalloproteinase 9 (MMP-9). MMP-9 is an extracellular operating Zn^2+^ dependent endopeptidase that is expressed in neurons and locally translated and released at the dendrites in response to enhanced neuronal activity driven by glutamate. MMP-9 plays an essential role in both establishing synaptic connections during development and in restructuring synaptic networks in the adult brain [147]. MMP-9 mRNA is part of the FMRP complex and localizes in dendrites. Translation of MMP-9 is increased at synapses in *Fmr1* KO mice suggesting its contribution to the aberrant dendritic spine morphology observed in the *Fmr1* KO mice and in FXS patients [148,149]. The genetic disruption of MMP-9 in the *Fmr1* KO mouse rescued key aspects of *Fmr1* abnormalities, including abnormal mGluR5-dependent LTD and dendritic spine abnormalities [150], providing evidence that MMP-9 is necessary to the development of FXS-associated defects in the *Fmr1* KO mouse. Interestingly, a high level of MMP-9 has been observed in the auditory cortex of adult *Fmr1* KO mice and the deletion of MMP-9 reversed the habituation defects [151]. A decreased MMP-9 activity in the hippocampus of the *Fmr1* KO mouse, dendritic spine maturation, improvement in anxiety, and strategic exploratory behavior were observed after treatment with the antibiotic minocycline [152]. These findings prompted the use of minocycline as a targeted treatment in humans with FXS through open-label trials which have demonstrated benefits with improvements in language, attention, social communication, and anxiety [153,154]. More recently, a controlled double-blind crossover study of minocycline for FXS treatment provided evidence for the safety of minocycline and showed benefits in global functioning in children with FXS [155]. In addition, as expected, the higher plasma activity of MMP-9 observed in FXS patients was lowered by minocycline in some patients [156], as minocycline is known to be a MMP-9 inhibitor [152]. On the other hand, no changes in plasma MMP 9 activity was found after treatment with sertraline [157], a selective serotonin-reuptake inhibitor which selectively blocks the uptake of serotonin at the presynaptic membrane, resulting in an increased synaptic concentration of serotonin in the central nervous system (CNS), and therefore, to an intensified serotonergic neurotransmission. Interestingly, a reduction of the MMP-9 levels was also reported in the *Fmr1* KO mouse following metformin treatment [139]. The results of the preclinical and clinical studies indicate that minocycline, through its mechanism of action as an MMP inhibitor, may be an additional potential effective avenue as FXS therapeutic treatment and MMP-9 activity, a potential biomarker in FXS.

## 10. Brain-Derived Neurotrophic Factor (BDNF)

Brain-derived neurotrophic factor is involved in the regulation of various processes of normal neural circuit function and development. Dysregulation in BDNF/TrkB signaling in the *Fmr1* KO mouse leads to altered brain development, including excessive sponginess, dendritic arborization [158], and impaired synaptic plasticity [159]. These neural alterations are promoted by activity-dependent variation in the sensitivity to BDNF-TrkB signaling, compensating postsynaptic activity [158].

The effects of reduced BDNF expression on the learning and behavioral phenotypes, including fear conditioning, pain behaviors, and hyperactivity, was examined in the *Fmr1* KO mouse crossed with a mouse carrying a deletion of one copy of the *Bdnf* gene (*Bdnf*+/−) [160]. The authors reported age-dependent alterations in the expression of BDNF in the hippocampus, reduced locomotor hyperactivity, deficits in sensorimotor learning, and startle responses typical of *Fmr1* KO mice. In addition, altered BDNF signaling in FMRP-deficient neural progenitor cells (NPCs) suggested that perturbations of brain development in FXS occur at very early stages of development [161].

A single-nucleotide polymorphism (SNP) in the human BDNF gene, leading to a methionine (Met) substitution for valine (Val) at amino acid 66, interferes with the intercellular trafficking and the activity-dependent secretion of BDNF in cortical neurons. One study found that the Val66Met BDNF polymorphism associates with epilepsy in a Finnish FXS male [162] but was not confirmed in a group of 77 patients with FXS [157]. However, a significant association between the BDNF polymorphism and improvements of several clinical measures was observed in a double-blind randomized placebo-controlled clinical trial of sertraline in FXS aimed to determine the efficacy of treatment in young children with FXS [157]. In addition, an open-label study showed a significant increase in BDNF level after treatment with the GABA_A_ agonist acamprosate [107]. Although more studies are warranted, these findings point to BDNF genotype as a potential molecular biomarker in FXS.

## 11. Amyloid-β Protein Precursor and Amyloid-β (APP, Aβ)

FMRP protein binds to the coding region of the APP mRNA and results in increased translation of the encoded product, the amyloid precursor protein (APP), which plays a vital role in the developing brain during synapse formation, while β-amyloid (Aβ) accumulation, results in synaptic loss and impaired neurotransmission. APP is processed by β- and γ-secretases to produce amyloid-β (Aβ), which is the prominent peptide found in the case of Alzheimer’s disease (AD).

A study by Westmark et al. found a 1.7-fold increase APP expression in the *Fmr1* KO mouse versus WT using western blot analysis and showed that the genetic knockdown of one APP allele in the *Fmr1* KO mouse rescued the FXS phenotypes including anxiety, seizures, mGLuR-LTD, and the ratio of mature versus immature dendritic spines [163]. APP and Aβ were evaluated as blood-based biomarkers and in a prospective open-label trial of acamprosate in pediatric subjects with FXS-associated autism spectrum disorder and found that acamprosate treatment significantly reduced sAPP and sAPPα [164].

Although blood levels of APP metabolites may not correlate with brain levels, which is one of the limitations of these studies, altogether these findings support a role for dysregulated APP production and processing in FXS and indicate that the APP metabolites may be viable biomarkers for FXS treatment.

## 12. Additional Potential Biomarkers

### 12.1. Ion Channels (CaV)

Voltage-gated ion channels are involved in neural transmission and some recent past studies showed their involvement in the FXS pathology [165]; more specifically, with the voltage-gated calcium channels (VGCC) family, namely Cav2.1 and Cav2.2 [166]. Synaptic transmission depends critically on presynaptic calcium entry via voltage-gated calcium (CaV) channels. FMRP regulates the expression of neuronal N-type CaV channels (CaV2.2) [166] and dysregulated calcium homeostasis, in addition to the decreased expression of the pore-forming subunit of the Cav2.1 channel, the Cacna1a gene, in *Fmr1*-KO cultured neurons [167]. Their findings indicate that FMRP plays a key role in calcium homeostasis during brain development; furthermore, the authors suggest that calcium homeostasis could be used as a cellular biomarker and for the identification of new drugs for target treatment in FXS.

### 12.2. Glycogen Synthase Kinase-3 (GSK-3)

Glycogen synthase kinase-3 (GSK-3) is a serine/threonine protein kinase that, when phosphorylated, regulates a variety of developmental processes, such as cell migration, cell morphology, neurogenesis, and gliogenesis via interaction with a variety of signaling pathways [168]. The lack of FMRP results in an abnormal increase in GSK-3β mRNA and protein levels in several regions of the brain [169] of the *Fmr1* KO mouse and in decreased hippocampal neurogenesis that likely contributes to the pathogenesis of FXS [170].

Several studies have demonstrated that lithium treatment rescued the FXS-associated impairments sustainable throughout the aging process in the Drosophila model of FXS [54,59]. In addition, GSK-3 inhibitors and lithium treatment provided the direct evidence of GSK-3 involvement in the pathology of FXS by reducing audiogenic seizure activity, improved performance on the open field elevated plus maze and passive avoidance tests [171], improved social defects [172], rescue of the hippocampus-dependent learning deficits [173], and improved cognitive deficits [174] in the *Fmr1* KO mouse. Additionally, the attenuation of reactive astrocytes, which has been observed in many brain regions of the *Fmr1* KO mouse with lithium treatment, provides further evidence of the involvement of GSK-3 in FXS [175]. These findings raise the possibility that GSK-3 activity may represent a biochemical mediator biomarker of impaired cognitive function in FXS and that modulators of its activity may have potential as therapeutic agents [176].

### 12.3. Striatal-Enriched Protein Tyrosine Phosphatase (STEP)

Striatal-enriched protein tyrosine phosphatase (STEP) is a brain-specific tyrosine phosphatase that plays a significant role in the development of the CNS by regulating dendritic proteins involved in synaptic plasticity [177,178]. STEP dysregulation is involved in the pathophysiology of several neuropsychiatric disorders [179], including FXS, likely by dephosphorylating both NMDARs and AMPARs [177]. While the enhanced activity of mGluRs in the absence of FMRP upregulates the translation of STEP [178,180,181] in the hippocampus of the *Fmr1* KO mouse, genetic reduction of STEP significantly diminishes some FXS-associated behaviors in *Fmr1* KO including seizures and restores select social and nonsocial anxiety-related behaviors [181]. Benzopentathiepin 8-(trifluoromethyl)-1,2,3,4,5-benzopentathiepin-6-amine hydrochloride (known as TC-2153) is a newly discovered STEP inhibitor [182]. A recent study [183] reported that this STEP inhibitor reduces seizure incidence and hyperactivity, anxiety and improves sociability, electrophysiological deficits in acute brain slices and spine morphology in *Fmr1* KO mouse. These observations suggest that STEP’s expression and activity could be useful for evaluating the clinical benefits of pharmacological therapeutic approaches in FXS targeting STEP.

### 12.4. Plasma Cytokines and Chemokines

Cytokines are the most important mediators of cell–cell communication in the human immune system. They perform a variety of functions like modulation of the central nervous system (CNS), brain functioning, and responses to infections or injury. Significant differences in plasma cytokine and chemokines levels were reported in patients with FXS including a high level of IL-1α, RANTES, and IP-10 [184]. It is currently unknown if the changes in the cytokine and chemokines are determinant in the development of FXS and if they occur throughout the lifetime of FXS patients, and therefore, their potential use as biomarkers needs more investigation.

### 12.5. Diacylglycerol Kinase Kappa (Dgkκ)

Diacylglycerol kinase kappa (Dgkκ) is a master regulator that controls two critical signaling pathways involved in protein synthesis. Lack of FMRP in the *Fmr1* KO mouse neurons results in the loss of Dgkκ expression along with mGluR1 receptor-dependent DGK activity, leading to synaptic plasticity alterations, dendritic spine abnormalities, and behavior disorders. These findings support the involvement of Dgkκ deregulation in FXS pathology and suggest that overexpression of Dgkκ in neurons could rescue the dendritic spine defects of the *Fmr1* KO mouse. Thus, DGKκ expression levels could represent a biomarker and targeting DGKκ signaling might provide new therapeutic approaches for FXS [185].

### 12.6. MicroRNAs (miRNAs)

MicroRNAs (miRNAs) are known as a class of small noncoding RNA molecules (19–23 nucleotides) that regulate almost 30% of genes at the post-transcriptional level in eukaryotic organisms [186]. Several studies have provided evidence of miRNA involvement in the pathogenesis of FXS by identifying and isolating several r(CGG)-derived miRNAs, including miR-fmr1-27 and miR-fmr1-42 in the zebrafish FXS model [187,188]. Their brain exhibits long dendrites and disconnected synapses, similar to those found in the human FXS hippocampal–neocortical junction [189]. Further, microarray analyses of miRNAs associated with FMRP in the *Fmr1* KO mouse brain identified miR-125a, miR-125b, and miR-132, and disruption of the regulating of these miRNA-mediated protein translation results in early neural development and synaptic physiology [190,191]. Another microarray study [192] in the *Fmr1* KO mouse showed the interaction of miR-34b, miR-340, and miR-148a with the Met 3′ UTR of the *FMR1* gene, suggesting that alterations in the miRNA expression resulted from the absence of FMRP, could contribute to the molecular pathology of FXS. Enhanced expression of miR-510, located on chromosome X in the 27.3Xq region, flanking to a fragile X site, was reported in full mutation female carriers [193]. Thus, although more studies are necessary to confirm their utility, many pieces of evidence indicate that miRNAs could be attractive candidate biomarkers in FXS.

## 13. Conclusions

FXS is a challenging disorder in terms of drug development and clinical implementation. An extensive preclinical work, carried out in the FXS animal models, has provided ways to improve the behavior, language, and cognitive ability, but several factors (complex clinical phenotype, genetic variability, gender differences, and use of multiple medications, limitations in the outcome measures and of tools) might have contributed to a lack of translation from the preclinical to clinical outcomes. When looking at the design of the preclinical studies to date, some limitations can be identified. Most of the FXS research in mammalian model systems is limited to two disease models, the *Fmr1* KO mouse and *Fmr1* KO drosophila animal model, but the central issue in using these models is variability and small effect size of the phenotype particularly in the area of cognitive defects. Moreover, overlapping phenotypes in these animal models sometimes may lead to over-prediction of the therapeutic potential of novel drug treatments.

Research to date on FXS has provided us with several potential candidate biomarkers that can, in principle, be used to assess efficacy; molecular biomarkers are promising, simple, and minimally invasive diagnostic tools that can objectively measure the biologically relevant effects of targeted treatments on the underlying molecular defects observed in FXS. However, the current research on molecular biomarkers in FXS suffers from a number of limitations. FXS is a neurological disorder, but brain tissue is not easily accessible. Therefore, biomarkers must be developed in a tissue that can be obtained easily, such as PBMCs, platelets, and fibroblasts. No single consistent molecule or modification state (i.e., phosphorylation or acetylation) has been reported to be differentially regulated in FXS patients versus controls consistently across multiple testing sites. Although many molecular biomarkers have been proposed in FXS (Figure 2), no one is accurate enough as changes according to disease modifications in a trial setting. No clinical history for any marker is available, and lengthy expensive processing and time consumption are required to generate test substrates such as primary fibroblasts (and induced pluripotent stem cells).

In summary, there is an urgent need to establish novel and reliable biomarkers in FXS, particularly blood-based biomarkers, essential to the development of new treatments. They can provide measures of disease severity and can be used to develop personalized treatments. Interestingly, when monitored over time, they can be used to evaluate treatment outcomes and help to identify responders, and therefore those individuals that following treatment have shown real benefit with phenotype improvements.

## Figures and Tables

**Figure 1 brainsci-09-00096-f001:**
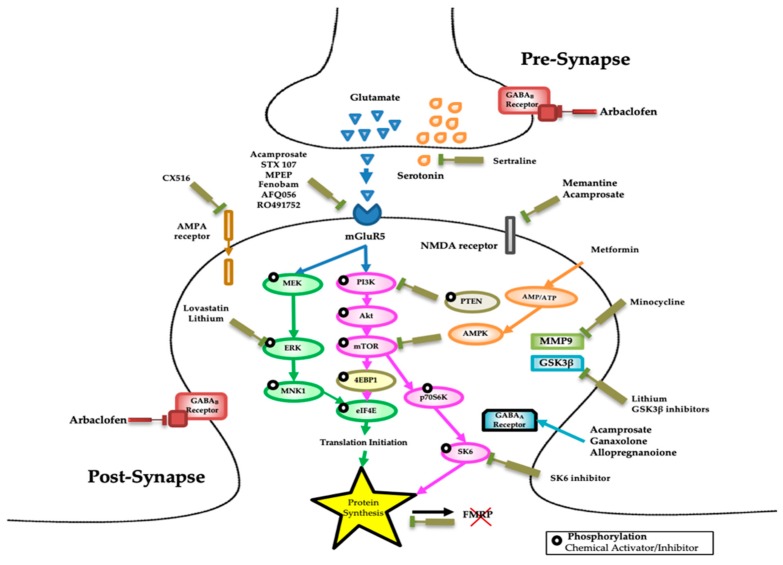
Potential therapeutic targets for fragile X syndrome (FXS). Diagram of the mechanisms implicated in FXS leading to altered synaptic plasticity. The figure also shows the molecular pathways targeted or understudy, for the reversal of cognitive and behavioral impairments in FXS patients. Several types of drugs, modulators, and compounds (inhibitor, agonist, and antagonist) can interfere with different pathways disturbed in FXS and have been used in a number of pharmacological treatments some of which are currently under investigation and are indicated in the figure.

**Figure 2 brainsci-09-00096-f002:**
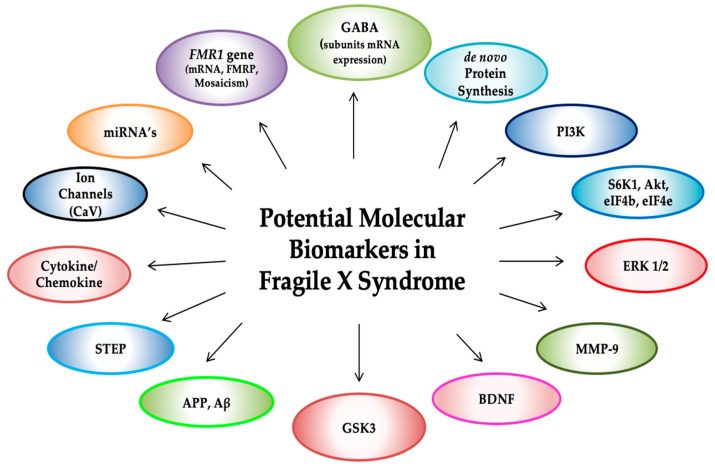
Candidate molecular biomarkers for FXS include a number of targets and substrates of several signaling pathways, in addition to fragile X mental retardation 1 (*FMR1*) molecular measures and metabolites, of which expression levels or activity have been found dysregulated in FXS animal models and in human FXS tissues. *Fmr1* mRNA and fragile X mental retardation 1 protein (FMRP) expression, de novo protein synthesis, γ-aminobutyric acid (GABA) receptors (GABA_A_ and GABA_B_), phosphoinositide 3-kinase (PI3K), extracellular-regulated kinase (ERK), matrix metalloproteinase-9 (MMP-9), brain-derived neurotrophic factor (BDNF), mammalian target of rapamycin (mTOR), p70 ribosomal S6 kinase (S6K1), ion channels (KNa, BKCa, CaV, Kv, HCN1), bone morphogenetic protein receptor Type 2 (BMPR2), Diacylglycerol Kinase Kappa (Dgkκ), endocannabinoid system (eCS), amyloid-β protein precursor (APP), microRNA’s (miRNA’s), striatal-enriched protein tyrosine phosphatase (STEP), glycogen synthase kinase-3 (GSK-3) cytokine and chemokine profiles, metabotropic glutamate receptor (mGluRs).

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
