# Peer review of "Molecular Biomarkers in Fragile X Syndrome"

_brainsci, 2019, doi:10.3390/brainsci9050096_

Reviewer 1 Report

The authors adequately addressed all my concerns and I consider the manuscript acceptable in its present form.

Author Response

Hi!!
Thank you so much for reviewing our draft and helping us in improving it.

Regards,

Marwa

Reviewer 2 Report

This manuscript is an extensive review of FXS itself, of drug treatments for FXS, and biomarkers for FXS.  The authors have done a great job in terms of comprehensiveness, on a topic that is crucial to the field, yet has been somewhat neglected in terms of a review on the subject. However, the paper could/should be improved by considering the points raised below.

1.    A shortcoming is that in some cases the incorrect or inappropriate references are cited.  The authors should double check that statements made in the text are backed up by the correct reference.  For example, in the second paragraph of the Introduction there is a statement that “…. 60% of those with the full mutation present with ASD [2].” Ref. [2} is a review paper on another topic and does not directly address the issue of incidence of ASD in the FXS population.  This needs to be rectified.  Also the figure 60% is much higher than cited in other papers.  The authors should go through the entire paper carefully to check this with all of the cited references to identify papers that best back up the statement being made. 

2.    The paper emphasizes biomarkers based on components of various signaling pathways, enzymes, and growth factors and cytokines. Absent is a discussion of other types of potential markers such as prepulse inhibition, EEG analysis, and others.  As such, the authors could consider changing the title to “Biochemical Markers in Fragile X Syndrome”.

3.    On lines 321-323 the authors make the statement that “The increased rCPS suggested an increase in the number of neurons, in the size of dendritic arborization and the number of spines in the mutant animals [110]. â€¨

What the Qin et al 2005 paper (ref. 110) says is “One interpretation of our results is that increased rCPS in Fmr1 null mice reflects an increase in the number of neurons and/or in the size of the dendritic arbor and number of spines in the mutant animals.”  This is not the same as demonstrating increased numbers of neurons. The authors should modify the text or back it up with more direct published evidence.

4.     In some places the text is awkward and/or redundant.  For example,

“Current non-pharmacological, behavioral and educational treatments are â€¨

100  symptomatic and extend to pharmacological treatments of symptoms, which include â€¨

101 anxiety, aggression and attention deficit hyperactivity disorder (ADHD). â€¨

            The English and sentence structure needs tweaking. Also, within the first few pages and sections of the paper there are several redundant statements, for example repetition of the symptoms of FXS as in this example.  Also, there are many places where the incorrect preposition or article was used.  The authors should go through the entire manuscript carefully to correct these grammatical errors. 

5.     A minor comment is that sometimes the authors use the term “a recent study…”. For example, A recent study [163]….” Reference 163 is from 2011, 8 year ago so is not really “recent”.  Again, checking throughout would be beneficial.

Author Response

Hi!!
I am attaching a cover letter enclosing all the point-by-point response to reviewer's comments.

Thanks,

Marwa

This manuscript is a resubmission of an earlier submission. The following is a list of the peer review reports and author responses from that submission.

Round  1

Reviewer 1 Report

This paper sets out to “review candidate molecular biomarkers that have been identified in the FXS mouse animal model and are now under validation for human applications or have already made their way to clinical trials”.   The issue of endpoints in clinical trials and the need for reliable biomarkers is much discussed in the field of research on fragile X syndrome.  Subjective or soft endpoints are thought to be a reason for large placebo effects in clinical trials. In this respect this review is timely.  The review is well organized and, for the most part, is comprehensive in citations.  The writing is uneven and there are some inaccuracies in the text as outlined below.

In parts of this manuscript there appears to be some confusion about the difference between a treatment target and a biomarker.  It seems that the bulk of this article is concerned with therapeutic targets.  If the purpose of the review is to report on potential biomarkers, it is appropriate to summarize therapeutic targets, but the substance of the article should address potential biomarkers.  It seems to me that the important issues are: 1.  Can the marker be measured reproducibly in human subjects?  2. Does it reflect disease severity? and 3. Does it change with efficacious treatment?  These questions should be addressed with respect to each potential biomarker.

The section on de novo protein synthesis is difficult to follow.  The authors discuss measurements made in hippocampal slices in vitro, brain in vivo and patient-derived cells in vitro with little explanation about the differences. In my opinion, putting these data together in a coherent manner with a discussion of the differences in approach is paramount in a review article. 

In the section on mTOR and S6K1, the emphasis on therapeutic targets despite opportunities to discuss measurements of markers in blood-derived cells.  The section on ERK did discuss the ERK activation assay in lymphocytes and measurements of ERK signaling in platelets as potential biomarkers, but the last paragraph reverts to a discussion of a potential therapeutic target (RSK) and refers to it as a candidate biomarker. 

There are numerous issues with respect to accurate representation of published studies and careful use of language. 

Some specific comments:

1.      Line 43.  FXS results in macroorchidism not microorchidism. 

2.      Line 77.  I don’t know of studies in human patients that show the electrophysiological characteristics of the Fmr1 KO mouse model.

3.      Lines 78-79.  The seizures in humans with FXS are spontaneous, whereas the seizures in Fmr1 KO mice are audiogenic.

4.      Lines 80-81.  I’m not sure what exactly the authors are referring to here.  Fmr1 KO mice have demonstrated deficits on reversal learning on the Morris water maze (18), acquisition of a visuospatial discrimination task (Krueger et al., 2010), trace conditioning (Guao et al., 2015), to name a few.  Not all labs find these differences in learning and memory, but many do report effects of the lack of FMRP on higher functions in mice and in drosophila (McBride et al., 2005; Bolduc et al., 2008). 

5.      Figure 1.  Lithium also acts on GSK3b.

6.      Figure 2.  How are these potential biomarkers?  How can they be applicable to clinical trials in human subjects?

7.      Lines 126-129.  Isn’t this overly simplistic?  It might depend on the AR in different brain regions, a variable not measurable in patients.

8.      Lines 131-132.  The data show that in Fmr1 KO mice it is mGluR-activated LTD that is excessive, not mGluR activation itself. 

9.      Lines 160-162.  This statement is not true.  There have been numerous studies showing that a single dose of Gp1 mGluR NAM affects the phenotype in Fmr1 KO mice.

10.  Lines 174-175.  Neither of the citations (49,66) report enhanced mGluR-dependent LTD in dfmr flies.

11.  Lines 228-231.  My reading of this paper by Braat et al., 2015, is that ganaxalone treatment reversed these phenotypes, not the cross with the YAC transgenic. 

12.  Lines 240-241.  I do not understand this sentence: “In a recent study, the response of the functional deficit FX neurons that lack the synaptic activity has been investigated to pulse application of the neurotransmitter GABA.”

13.  Lines 253-255.  There’s something wrong with this sentence: “Indeed, two phase3 placebo-controlled trials conducted with a flexible dose trial in subjects age 12-50 in subjects age 5-11 to determine the safety and efficacy of arbaclofen for social avoidance in FXS patients.”

14.  Lines 257-258. There’s something wrong with this sentence: “Later, in a double-blind, placebo-controlled crossover trial of GABAB agonist baclofen [98] improved the social avoidance in FXS patients.”

15.  Lines 269-271.  There’s something wrong with this sentence: “These preclinical and clinical studies strengthen the hypothesis of GABA receptors involved in the pathology of FXS and point GABA modulators as potential biomarkers in target treatment in FXS.”  How can GABA receptor modulators be biomarkers?

16.  Line 285-286.  The citation (104) does not report an increase in number of neurons, size of dendritic arborization and number of spines.

17.  Lines 335-337.  This sentence does not make sense.  I see how a p110b-selective antagonist can be a potential therapeutic, but I don’t understand how it can also be a biomarker.

18.  Lines 355-360.  These studies were in lymphocytes, not fibroblasts.

19.  Lines 374-375.  In reference [131] there is no work on MMP9. 

Minor concerns:

1.      Lines 30-38.  The first paragraph defines a biomarker in the first sentence and then reiterates the definition in other words in the next three sentences. 

2.      Line 39.  FXS is and X-linked disorder and the most prevalent inherited cause of intellectual disability.

3.      Lines 149-150.  This reference to the FRAXAD mouse model seems totally out of place here and may be confusing to the reader. In addition, FRAXAD is not defined.

4.      Line 150. MPEP is not defined.

5.      Lines 113-116.  If this statement is true, please provide a reference. 

6.      Line 262.  Please substitute the word children for kids.

7.      Line 328. Gp1 is not defined.

8.      Line 339 and 344.  mTOR is defined twice.

Reviewer 2 Report

This is an interesting review, addressing the importance of identifying measurable biomarkers for Fragile X Syndrome (FXS). The authors present a very detailed review of the available literature, describing in detail a number of biomarkers for the disease. As it is written the review may certainly be of interest for a broad readership, but it lacks some speculation about novel strategies to identify novel potential FXS biomarkers. The problem in FXS is that, as also reported in this review, most of the drugs tested so far failed to really improve the multiple disease symptoms. Thus, identification of novel therapeutic strategies is crucial and the authors should better speculate on this point.

In addition the authors should correct their statement at lines 221-224. There are 3 types of GABA receptors (A,B, C), not 2 as stated. Please correct.

Finally, the manuscript requires a careful and thorough editing for minor but consistent language and typographical errors. As an example, the use of commas seems not accurate. I try to report here some of this errors, but my list is not complete.

Line 48: which function > whose function

Line 57: have also observed > have also been observed

Line 70: understudy > under study

Line 78: show the patter > show pattern

Lines 98-106: Figure 2 legend should at least report a brief description of the figure, and not a simple list.

Lines 120-122: the whole sentence is not clear, please rephrase

Line 141: to check the hypethesis that unchecked…rephrase to avoid repetition

Lines 208-210 the verb is missing, please rephrase

Line 303: have observed > have been observed

Lines 396-399: the whole sentence is not clear, please rephrase

Lines 428-431: the whole sentence is not clear, please rephrase

Lines 464-465: “BDNF…male” > the whole sentence is not clear (what does “finish” mean? Maybe “Finnish”?). Please rephrase

Lines 501-502: the whole sentence is not clear, please rephrase (it does not make sense to say that “the sodium activated potassium channel is the largest known potassium channel subunit”…)

Line 611: mirna’s > mirnas

Reviewer 3 Report

A review “Molecular biomarkers in Fragile X Syndrome” by Zafarullah and Tassone is a very comprehensive description of current status on the research related to FXS biomarkers and treatments.

In general this contribution well describes current status of the FXS field. However, there are several aspects of this manuscript that requires an improvement.

1. A more detailed description of molecular defect, genetics, repeat sizes with mentioning permutation related diseases would be of benefit to a broader audience not familiar with FXS.

2. Figure 2 – it is an exact recapitulation of biomarkers/therapeutic approaches described in the main text and as such it is redundant (looking at the figure does not bring any more information than simply reading the texts). I recommend removing this figure. I think it would be of benefit to present a comprehensive figure that would try to classify biomarkers and treatments into a few categories and connected them with particular molecular or physiological changes in FXS.

3. The title of the contribution suggests a focus on molecular biomarkers. However, reading some of the subchapters it appears that the focus is on the treatment approaches. For example, fragment about mTOR- it is not clear how this pathway is a potential biomarker. Perhaps it would be of benefit to the reader to classify presented biomarkers into generally accepted categories (e.g. predictive, diagnostic, surrogate end point etc.) and present them in the context of FXS biomarkers as a figure or a table.

4. The entire article needs to be carefully edited for English and spelling. A fe examples are given below (it is not a comprehensive list, entire paper should be edited):

Lane 218 – 219 grammar, line 343 lack of ].; lane 355 grammar “rate initiation rates”; lane 372 lack of ]; lane 374 Metformin is sometimes capitalized and sometimes not, unify.; lane 375 lack of ]; lane 378 lack of ]; lane 420 Glutamate does not need to be capitalized; lane 430 “of” instead of “or”; lane 462 Methionine and Valine do not need to be capitalized; lane 464 “finish” should be “Finish”; lane 646 remove underline; lanes 661 -662 this sentence is not clear; lane 665 what is “stable biomarker” please define or remove; lane 665 – 669 –sentence is not clear/grammar sentence is extremely long; etc. etc.